# Theranostic Agent Combining Fullerene Nanocrystals and Gold Nanoparticles for Photoacoustic Imaging and Photothermal Therapy

**DOI:** 10.3390/ijms23094686

**Published:** 2022-04-23

**Authors:** Riku Kawasaki, Kosuke Kondo, Risako Miura, Keita Yamana, Hinata Isozaki, Risako Shimada, Shogo Kawamura, Hidetoshi Hirano, Tomoki Nishimura, Naoki Tarutani, Kiyofumi Katagiri, Alexandra Stubelius, Shin-ichi Sawada, Yoshihiro Sasaki, Kazunari Akiyoshi, Atsushi Ikeda

**Affiliations:** 1Applied Chemistry Program, Graduate School of Advanced Science and Engineering, Hiroshima University, 1-4-1 Kagamiyama, Higashi-Hiroshima 739-8527, Japan; riku0528@hiroshima-u.ac.jp (R.K.); konstarkk3033@icloud.com (K.K.); d216326@hiroshima-u.ac.jp (K.Y.); m211491@hiroshima-u.ac.jp (H.I.); rishima0624@gmail.com (R.S.); m213831@hiroshima-u.ac.jp (S.K.); m224960@hiroshima-u.ac.jp (H.H.); n-tarutani@hiroshima-u.ac.jp (N.T.); kktgr@hiroshima-u.ac.jp (K.K.); 2Department of Energy and Hydrocarbon Chemistry, Graduate School of Engineering, Kyoto University, Kyoto Daigaku Katsura, Nishikyo-ku, Kyoto City 615-8510, Japan; 3Department of Chemistry and Materials, Faculty of Textile Science and Technology, Shinshu University, 3-15-1 Tokida, Ueda 386-8567, Japan; nishimura_tomoki@shinshu-u.ac.jp; 4Division of Chemical Biology, Department of Biology and Biological Engineering, Chalmers University of Technology, SE-412 96 Gothenburg, Sweden; alexandra.stubelius@chalmers.se; 5Department of Polymer Chemistry, Graduate School of Engineering, Kyoto University, Kyoto Daigaku Katsura, Nishikyo-ku, Kyoto City 615-8510, Japan; sawada.shinichi.4u@kyoto-u.ac.jp (S.-i.S.); sasaki.yoshihiro.8s@kyoto-u.ac.jp (Y.S.); akiyoshi.kazunari.2e@kyoto-u.ac.jp (K.A.)

**Keywords:** fullerene nanocrystal, gold nanoparticle, photoacoustic imaging, organic–inorganic hybrid nanoparticle

## Abstract

Developing photoactivatable theranostic platforms with integrated functionalities of biocompatibility, targeting, imaging contrast, and therapy is a promising approach for cancer diagnosis and therapy. Here, we report a theranostic agent based on a hybrid nanoparticle comprising fullerene nanocrystals and gold nanoparticles (FGNPs) for photoacoustic imaging and photothermal therapy. Compared to gold nanoparticles and fullerene crystals, FGNPs exhibited stronger photoacoustic signals and photothermal heating characteristics by irradiating light with an optimal wavelength. Our studies demonstrated that FGNPs could kill cancer cells due to their photothermal heating characteristics in vitro. Moreover, FGNPs that are accumulated in tumor tissue via the enhanced permeation and retention effect can visualize tumor tissue due to their photoacoustic signal in tumor xenograft model mice. The theranostic agent with FGNPs shows promise for cancer therapy.

## 1. Introduction

Theranostic nanomaterials are one of the most promising candidates in precise cancer therapy for their excellent predictability of therapeutic efficacy by coupling diagnostic imaging with therapy in one system [1,2,3]. Light-triggered theranostic agents are advantageous in achieving cancer treatment due to their noninvasiveness, spatiotemporal addressability, and efficient therapeutic efficacy based on photothermal therapy (PTT) [4] and photodynamic therapy toward solid tumors (PDT) [5].

With high spatial resolution, penetrability with deeper tissue, and noninvasiveness, photoacoustic imaging (PAI) is a promising biomedical imaging modality to visualize disease targets [6]. PAI exploits the PA effect that is conversion of the energy from irradiated light into ultrasonic [7]. The ultrasonic signal shows several advantageous points, including negligible scattering and dissipation in biological tissues, and PAI can provide excellent contrast compared to optical imaging based on fluorescence [8].

To exploit the availability of PAI as a diagnosis modality, various types of contrast agents have been developed, such as gold nanocrystals [9,10,11], fluorophores [12,13,14], and nanocarbons [15,16,17]. With excellent absorbability in the visible light (>600 nm) and near-infrared regions, which can reach deeper tissues in the body, nanocarbons [18], including carbon nanotubes [15] and graphene [19], have been used for PAI. Despite the high efficiency of intersystem crossing, the use of fullerenes for PAI has been limited due to their native hydrophobicity [15] and low absorbability in the near-infrared red (NIR) region [17].

We have addressed the water solubilization of hydrophobic π-conjugated molecules including fullerenes [20,21], porphyrins [22,23], chlorins [24], and phthalocyanines [25] using biomolecules. Moreover, we investigated dyad coupled fullerene systems to expand the absorbability of fullerenes [21]. Recently, we developed a water-dispersible nanoaggregate comprising fullerenes, that is, fullerene nanocrystals (FNCs), via supramolecular chemistry techniques [26]. Hybridization of FNCs with fluorophores such as porphyrins enabled to expand their absorbability in the visible light (>620 nm) and NIR regions [27]. Moreover, fullerene molecules have high crystallinity in FNCs and their highly packing state in each nanoparticle are advantageous for both photoacoustic signal enhancement in PAI and efficient light-to-heat energy conversion in PTT. [28] These features of FNCs encouraged us to develop FNC-based theranostic nanomaterials coupled with gold nanoparticles (GNPs). Here, GNPs are also used as theranostic reagents due to their excellent surface plasmon resonance [29], and GNPs activated by photoirradiation in the region at visible light can provide energy to fullerene [30,31,32]. From this point of view, hybridization of FNCs with GNPs will exhibit synergistic effects by expanding the absorbability of FNCs in visible light via energy transfer from FNCs to GNPs.

In this study, we demonstrated PAI-based tumor imaging in vivo and PTT in vitro using hybrid nanoparticles, coupling FNCs with GNPs (FGNPs) (Figure 1). By exposing them to photoirradiation with optimal wavelength (>620 nm), FGNPs could efficiently convert energy from light into ultrasounds and thermal energy compared to FNCs and GNPs, which makes them potentially applicable as a theranostic agent for PA and PTT. We evaluated the PTT effects of FGNPs against a murine colon carcinoma cell line (Colon26). Our hybrid nanoparticles enhanced photo-induced cytotoxicity against Colon26 cells and strong PA signals were detected within cells. In addition, results from tumor tissue imaging by PAI in tumor xenograft mice suggested that our system was more practicable as a contrast agent for PA-based cancer diagnosis.

## 2. Results and Discussion

The formulation of FNCs was carried out as in previously established methods. Afterward, GNPs were ripened in the presence of citric acid in the resulting dispersion (FNCs, 1.0 mM; Au^3+^, 1.0 mM; citric acid, 38.8 mM) [33]. As a control, we additionally formulated GNPs via same method. The formulation of GNPs, FNCs, and FGNPs were confirmed by measuring the UV-Vis absorption spectra. As shown in Figure 2a, absorption spectra from fullerene became broadened in comparison with the C_60_@γ-CD complex after the heating process, indicating fullerene became aggregated during the procedures [26]. After the ripening of GNPs in the presence of FNCs, a peak around 530 nm was found, which is characteristic of GNPs, suggesting GNPs can be formed on the surface of FNCs. To confirm the morphology of GNPs, FNCs, and FGNPs, we carried out transmission electron microscopy (TEM). The morphologies of FNCs were irregular with a diameter of 50–80 nm (Appendix A), and round shapes with a diameter of 15–20 nm were found in the case of GNPs (Appendix A), as we previously reported. As shown in Figure 2b, GNPs with a diameter of 13 nm with relatively narrow dispersity in their size were absorbed on the surface of the FNCs. In addition, individual GNPs and FNCs were not found in the current condition, suggesting the hybrid nanoparticles of FNCs with GNPs were successfully prepared, and similar morphology was found in the case of the field emissive-scanning electron microscope (Appendix A). The absorption of GNPs on FNCs is advantageous in working as light-harvesting antenna units. Dynamic light scattering (DLS) measurement revealed that the hydrodynamic diameters of FNCs, GNPs, and FGNPs are 84, 13, and 98 nm, respectively (Table 1 and Appendix A). All these values correspond to the TEM images, and their size were reasonable for passive tumor targeting property, that is, enhanced permeable retention (EPR) effects. In addition, all these nanoparticles were negatively charged in the current condition (pH, 7.4; 25 °C) due to electrochemical properties of citric acid and fullerenes. Induced coupled plasma (ICP) spectroscopy revealed the concentration of gold in prepared GNPs and FGNPs to be 300 ppm and 313 ppm, respectively. This suggests that FNCs do not prevent gold nanoparticles from ripening in the current condition.

Crystallinity of the molecules in the probes is critical in enhancing PA signals and photothermal heat characteristics because the energy from irradiated light can be efficiently converted into ultrasounds and/or heat in the case of crystal state and aggregation form. [28] To address the crystallinity of fullerenes in each nanoparticle, we carried out wide angle scattering measurement (WAXS) of FNCs and FGNPs in SPring-8 (BL40B2; camera length, 0.098 m). Peaks from the crystal of fullerene were assigned in FNCs (Figure 2c, red line). In addition, the diffraction peaks from FGNPs were slightly shifted from FNCs (Figure 2c, blue line), indicating crystal structure of fullerenes slightly changed during the ripening processes. The WAXS measurements suggest that fullerenes are highly packed in FNCs and FGNPs by forming crystal structures, and the hybrid systems have great potential as PAI contrast agents and PTT agents.

Thermogravimetric (TG) analyses were carried out to estimate the weight of the component in FGNPs. FGNPs were heated from room temperature up to 900 °C, with a heating rate of 5 °C·min^−^^1^. As shown in Figure 2d, the weight loss of FGNPs at 40−180 °C, 200−400 °C, and 600−850 °C represents the removal of water, combustion of citric acid [34], and combustion of fullerene [35], respectively. The total TG weight loss of FGNPs in the above temperature range was ~90 wt%, suggesting that FGNPs contain metallic gold of approximately 10 wt%. These results supported that FGNPs contain both fullerene and citric acid coated GNPs.

We next conducted sustainability and stability of these nanoparticles (FNCs and FGNPs) in aqueous media by measuring the UV-Vis absorption spectra and DLS. For one-week incubation, no apparent changes in absorption spectra were found in all these systems (Appendix A). In addition, the hydrodynamic diameter and PDI value of the FNCs and FGNPs did not change during the period, suggesting these systems were also colloidally stable in aqueous media (Appendix A). To evaluate availability in biologics, we further examined the stability of FNCs, GNPs, and FGNPs in cell culture media. In the case of FGNPs, the hydrodynamic diameter increased from 98 nm to 150 nm, indicating the absorption of serum proteins resulting in secondary aggregation of FGNPs (Appendix A). Though FGNPs formed secondary aggregation in culture media conditions, their hydrodynamic diameter did not change after the formulation of secondary aggregation with maintaining the particle size corresponding to EPR effect, suggesting that our system can be potentially applicable for a contrast agent for tumor visualization.

We measured PA signals from GNPs, FNCs, and FGNPs (C_60_, 1 mM; Au, 300 ppm) with light irradiation at 680 nm, which can penetrate deeply into tissues. The PA signals from GNPs, FNCs, and FGNPs were quantified to be 2.5 × 10^6^, 1.7 × 10^7^, and 8.2 × 10^7^, respectively (Figure 2e). FNCs and FGNPs exhibited stronger PA signals than GNPs, suggesting a highly packed state of fullerenes with crystal regions in FNCs, and that FGNPs are advantageous in converting energy from light into ultrasounds as we expected. In addition, the PA signals from FGNPs were 4.8 times higher than that from FNCs. The phenomena should be caused by energy transfer to FNCs from GNPs [30,31,32].

To address the application for photothermal therapy, we conducted photothermal heating characteristics of FNCs, GNPs, and FGNPs by exposing them light with an optimal wavelength (>620 nm). Dispersion of FNCs, GNPs, and FGNPs were irradiated for 15 min (C_60_, 1 mM; Au, 300 ppm) and changes in temperature were monitored by digital thermometer (Figure 2f). In absence of nanoparticles, the temperature of solutions did not change during the period. In contrast, photothermal heating effect was found in the case of FNCs, GNPs, and FGNPs. In addition, the photothermal conversion efficiency of FGNPs was the highest among these three agents, and the dispersion of FGNPs showed an approximate 10 °C increase after 15 min irradiation. The results are comparable to results in PA signals. For these results, we conducted the following studies using FGNPs.

To address the availability of FGNPs as a theranostic agent for PTT and PAI, we initially conducted cytotoxicity tests against a murine fibroblast-like cell line (L929) and murine colon carcinoma line (Colon26). After 24 h incubation with FGNPs, no apparent cytotoxicity was found even at the highest concentration (C_60_, 0.1 mM; Au, 30 ppm) against both cell lines (Figure 3a), indicating FGNPs are non-toxic theranostic agents.

We next investigated the interaction of the hybrid systems with Colon26 with varying the concentration of FGNPs (Au, 0.1, 1, and 10 ppm) by quantifying the cellular uptake amount of gold using ICP-AES (Figure 3b). With increasing the concentration of FGNPs, larger amounts of the hybrid nanoparticles were accumulated in Colon26 cells, suggesting cancer cells can be efficiently visualized by PAI and killed by PTT via photoirradiation.

To address application for PTT, photoirradiation (>620 nm) was carried out on Colon26 cells treated with FNCs, FGNPs, or GNPs. Cell destruction was dose-dependently induced to the cells in FGNPs and GNPs (Figure 3c), but FNCs could not kill cancer cells even at the highest concentration (1 mM of C_60_) (Appendix A). The 50% inhibition concentration (IC_50_) value using GNPs and FGNPs was determined to be 104 and 1.1 ppm, respectively. FGNPs enhanced photo-induced cytotoxicity by hybridization of GNPs with FNCs. We next measured PA signals within Colon26 cells by excitation with a 680 nm laser. The strong photoacoustic signals from FGNPs were detected within cells (Figure 3d). These results suggested that FGNPs are potentially applicable as theranostic agents for PA and PTT.

We finally demonstrated the visualization of tumor tissue in tumor xenograft mice using FGNPs. Tumor xenograft mice were established by implantation of Colon26 cells (5.0 × 10^5^ cells) in the back of Balb/c mice (4-week-old, female, 18 g). After the tumor grew up to 5 mm in size, dispersion of FGNPs was administrated intravenously (10 ppm, 100 μL). At each time point (3, 6, and 24 h), blood and tumor tissue were collected from the treated mice to evaluate blood retention and tumor accumulation by quantifying the amount of gold using ICP-AES (Figure 4a). The concentration of gold in the blood stream gradually decreased with time. In contrast, the accumulation of FGNPs in tumors increased with time, and FGNPs were accumulated at 1.8 ppm in tumor tissue at 24 h. This accumulation should be achieved by mainly via EPR effect.

The accumulation of FGNPs in tumor tissue was visualized by PAI. Compared to the PBS injected group as the nontreated control, stronger PA signals were detected in tumor tissue in FGNPs treated group (Figure 4b,c), and the PA signals from FGNPs gradually increased with time (Figure 4d). This result is comparable to the accumulation of biodistribution study. For these results, FGNPs were accumulated in tumor tissue mainly via the EPR effect and the accumulated FGNPs could successfully visualize tumor tissue in vivo by PAI, suggesting our hybrid system is practical as a theranostic agent based on PAI and PTT.

## 3. Conclusions

Fullerene nanocrystals formed a stable hybrid with gold nanoparticles (FGNPs). For high crystallinity and energy transfer from gold nanoparticles to fullerene crystals, the FGNPs exhibited stronger photoacoustic signals and photothermal heating character than gold nanoparticles and fullerene crystals by applying light with an optimal wavelength. FGNPs exhibited potential for photothermal therapy in vitro. Furthermore, FGNP systems efficiently accumulated in tumor tissue mainly via the EPR effect, and the FGNPs in the tumor could visualize tumor tissue by photoacoustic imaging. This platform may be an excellent candidate for a theranostic agent based on photoacoustic imaging and photothermal therapy. The absorbability of our system might be insufficient to visualize specific organ and disease, because the absorption in visible light is partially overlapped with the heme proteins such as myoglobin and hemoglobin. From this point of view, ripening of the gold nanorod on the surface of FNCs can exploit the absorbability in the NIR region. Moreover, the modification of pilot molecules on the surface of FGNPs opens the opportunity for the application to visualize specific disease with high sensitivity.

## 4. Materials and Methods

### 4.1. Materials

Fullerene was purchased from Frontier Carbon (Tokyo, Japan) and C_60_ was used after purification with extraction and silica gel column chromatography. γ-cyclodextrin (γ-CDx), sodium citrate, and HAuCl_4_ were obtained from Fujifilm-Wako pure chemical industries, Ltd. (Tokyo, Japan). Polyethylene glycol monomethyl ether (PEG, *Mw* = 2000) was purchased from Sigma-Aldrich (St. Louis, MO, USA). Colon26 cells were maintained in Dulbecco’s Modified Eagle Medium (DMEM) containing 10% fetal bovine serum (FBS) and 1% penicillin-streptomycin (PS) and L929 cells were maintained in DMEM containing 10% FBS, 1% glutamic acid, and 1% PS. Balb/c mouse (4-week-old, male) were purchased from Japan SLC (Shizuoka, Japan). Tumor xenograft mice were established by subcutaneous injection of suspension of Colon26 cells (5.0 × 10^5^ cells/50 μL) toward the back of Balb/c mouse. The animal experiments were performed in accordance with the Guidelines for Care and Use of Laboratory Animals of Hiroshima University and were approved by the Ethics Committee for Animal Welfare of Hiroshima University (accreditation, C21-34).

### 4.2. Preparation of the C_60_@γ-CD Complex

C_60_ (5.0 mg, 6.9 × 10^−6^ mol) and γ-CDx (36 mg, 2.8 × 10^−5^ mol) were placed in an agate capsule containing two agate mixing balls. The mixture was mashed vigorously at 30 Hz for 20 min using a high-speed vibration mill (MM200; Retcsch Co., Haan, Germany). The resulting mixture was extracted with deionized water (2 mL) and precipitation was removed by centrifugation (14,000 ppm, 20 min) and filtration (0.45 μm).

### 4.3. Preparation of Fullerene Nanocrystal (FNCs)

C_60_@γ-CDx complex (C_60_, 1 mM; γ-CDx, 2 mM; 0.2 mL) was interacted with PEG (0.2 mL, 50 g·L^−1^) and water (1.6 mL) was added, and the resulting solution was heated at 80 °C for an hour. The formulation of FNCs was confirmed by measuring UV-Vis absorption spectra (3600 UV-vis-NIR spectrometer, Shimadzu, Tokyo, Japan), dynamic light scattering (Zeta-sizer Nano, Malvern, UK), and transmission electron microscope images (JEM-1400, JEOL Ltd., Tokyo, Japan).

### 4.4. Preparation of Fullerene Nanocrystal/Gold Nanoparticle Hybrid (FGNPs)

HAuCl_4_ (1 mM, 2 mL) was added to dispersion of FNCs (C_60_, 1 mM; γ-CDx, 2 mM; 1 mL) and sodium citrate solution (38.8 mM, 10 mL) was injected dropwise. The resulting dispersion was stirred for 30 min with heating. After cooling down, FGNPs were isolated by centrifugation (1500 rpm, 20 min). The formulation of FNCs was confirmed by measuring UV-Vis absorption spectra, dynamic light scattering, and transmission electron microscope images.

### 4.5. Wide Angle X-ray Scattering Measurement

WAXS measurements were performed at BL40B2 of SPring-8, Japan. A 7.73 × 3.86 cm^2^ photon-counting detector (EIGER 2 S) was placed 0.098 m away from the sample. The wavelength of the incident beam was 0.1 nm. This setup provided a *q* range of 2–30 nm^−1^, where *q* is the magnitude of the scattering vector defined as the following equation.
*q* = 4πsin θ/λ

The measurement was carried out at 25 °C with an exposure time of 180 s. The results were converted to with the software package FIT2D.

### 4.6. Photothermal Heating Character

Dispersion of FNCs, GNPs, and FGNPs (C_60_, 1 mM; gold, 300 ppm) was irradiated with light (15 W·cm^−2^, >620 nm) using Xenon lamp (SX-UID500X, Ushio Inc., Tokyo, Japan) equipped with long path filter and the temperature of dispersion was measured at each time point (0, 1, 2, 3, 4, 5, 10, and 15 min) by digital thermometer (AD-5636, ASONE, Osaka, Japan).

### 4.7. Photoacoustic Signal of Dispersion

Dispersion of FNCs, GNPs, and FGNPs (C_60_, 1 mM; gold, 300 ppm) was excited with pulse laser (680 nm) and the photoacoustic signals were detected with Nexus128 (ENDRA).

### 4.8. Cell Viability Assay

L929 or Colon26 cells were seeded on a 96-well plant at a density of 5.0 × 10^3^ cells/well and incubated overnight. The cells were exposed to FNCs, GNPs, and FGNPs with varying concentration for 24 h. In the control, we added same volume of MilliQ (1 μL), which is used to prepare samples, to the cells. After addition of Cell Counting Kit-8, the cell viability was quantified by measuring absorbance at 450 nm and 650 nm (reference wavelength) using microplate reader (MPR-A100, ASONE).

### 4.9. Photothermal Therapy In Vitro

Colon26 cells were seeded on a 96-well plate at a density of 5.0 × 10^3^ cells/well and incubated overnight. The cells were exposed to FNCs, GNPs, and FGNPs with varying concentration for 24 h. The cells were irradiated for 30 min (>620 nm) using Xenon lamp. After addition of Cell Counting Kit-8, the cell viability was quantified by measuring absorbance at 450 nm and 650 nm (reference wavelength) using microplate reader.

### 4.10. Cellular Uptake and Photoacoustic Imaging In Vitro

Colon26 cells were seeded on a 12-well plate at a density of 1.0 × 10^5^ cells/well and incubated overnight. The cells were exposed to FGNPs with varying concentration for 24 h and the cells were collected with trypsin. The cells were digested with aqua regia and accumulation within cells was quantified with ICP-AES. In the case with photoacoustic imaging, the isolated cells were excited at 680 nm using Nexus128.

### 4.11. Biodistribution of FGNPs

Tumor xenograft model mouse was established by subcutaneous injection of suspension of Colon26 cells (5.0 × 10^5^ cells/50 μL). After 10 days of incubation, dispersion of FGNPs in PBS was administrated via intravenous injection (10 ppm, 100 μL). Tumor tissue and blood was isolated at each time point (3, 6, and 24 h) and collected organs were digested with aqua regia. The resulting solution was analyzed by ICP-AES. Here, the tumor volume was calculated by following equation.
Tumor volume = (long axis of tumor) × (short axis of tumor)^2^/2

### 4.12. Photoacoustic Imaging In Vivo

Tumor xenograft model mouse was established by subcutaneous injection of suspension of Colon26 cells (5.0 × 10^5^ cells/50 μL). After 10 days of incubation, dispersion of FGNPs in PBS were administrated via intravenous injection (300 ppm, 100 μL). Tumor tissue was visualized by Nexus128 at each time point (3 and 24 h).

## Figures and Tables

**Figure 1 ijms-23-04686-f001:**
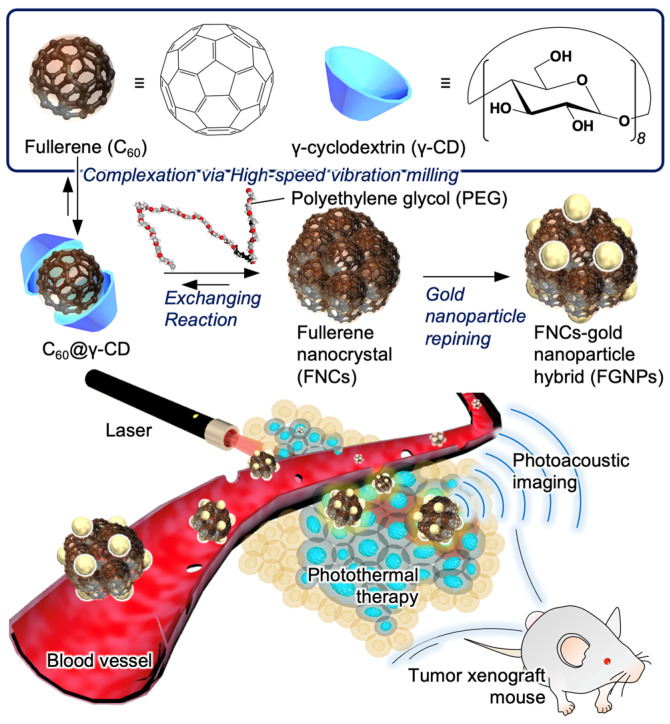
Preparation of hybrid of fullerene nanoparticles with gold nanoparticles (FGNPs) and tumor visualization with FGNPs by photoacoustic imaging.

**Figure 2 ijms-23-04686-f002:**
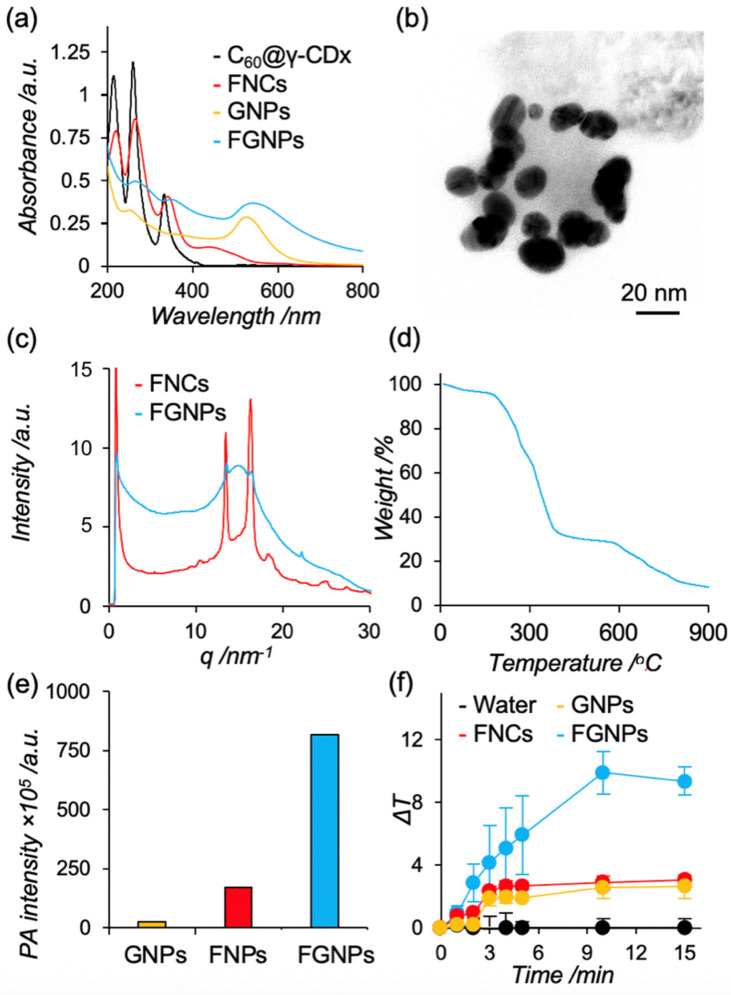
Basic characterization of FGNPs. (**a**) UV-Vis absorption spectra of C_60_/γ-CD (black), FNPs (red), GNPs (yellow), and FGNPs (blue) (C_60_, 1 mM; γ-CD, 2 mM; gold, 300 ppm). (**b**) Representative morphology of FGNPs. The samples were observed without staining by TEM (accelerate voltage, 80 kV). The scale bar represents 20 nm. (**c**) The 1D WAXS intensity profiles of FNPs (red) and FGNPs (blue). The detector was placed 0.098 m away from samples. The wavelength of the incident beam was 0.1 nm. The measurement was carried out at 25 °C with an exposure time of 180 s. (**d**) Thermogravimetric curve of FGNPs. (**e**) Photoacoustic signals of GNPs (yellow), FNPs (red), and FGNPs (blue). The samples were excited at 680 nm (C_60_, 1 mM; Gold, 300 ppm). (**f**) Photothermal heating characteristics of FNPs, GNPs, and FGNPs. Water (black) and dispersion of FNPs (red), GNPs (yellow), and FGNPs (blue) were exposed to light with optimal wavelength (>620 nm, 5 W·cm^−2^) (C_60_, 1 mM; Gold, 300 ppm).

**Figure 3 ijms-23-04686-f003:**
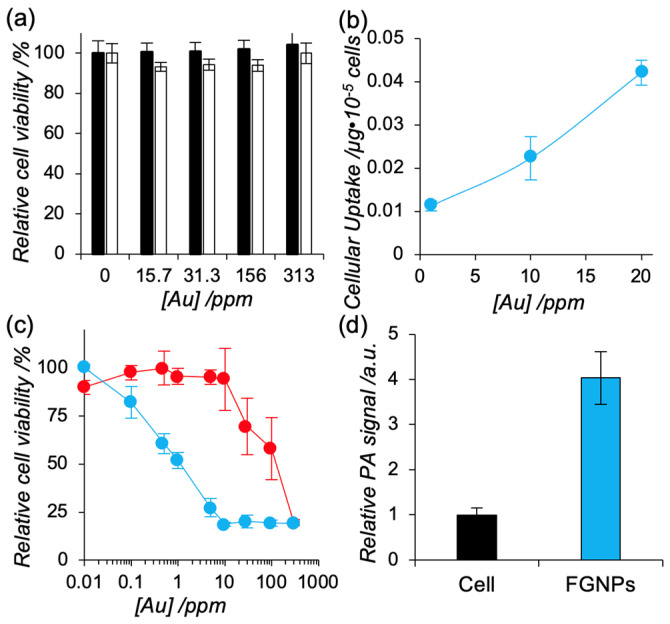
(**a**) Cytotoxicity of FGNPs against murine fibroblast (L929, black) and murine colon carcinoma (Colon26, white). The cells were co-incubated with FGNPs with varying concentration for 24 h. Cell viability of the treated cells were quantified with Cell Counting Kit-8 (*n* = 3). Data represent mean ± SD. (**b**) Cellular uptake amount of FGNPs toward Colon26. Colon26 cells were exposed to FGNPs with varying concentration of FGNPs for 24 h. Concentration of gold within cells was quantified by ICP-AES (*n* = 3). Data represent mean ± SD. (**c**) Photo-induced cytotoxicity toward Colon26 cells using FGNPs (blue) and GNPs (red). The cells were co-incubated with FGNPs with varying concentration for 24 h. The cells were exposed to light with optimal wavelength (>620 nm) for 30 mins. After 24 h incubation, the cell viability was confirmed by Cell Counting Kit-8 (*n* = 3). Data represent mean ± SD. (**d**) Photoacoustic signal from Colon26 cells treated with FGNPs (blue). Colon26 cells were treated with FGNPs (20 ppm, blue) or MilliQ (nontreatment control, black) for 24 h and the collected cells were excited at 680 nm (*n* = 3). Data represent mean ± SD.

**Figure 4 ijms-23-04686-f004:**
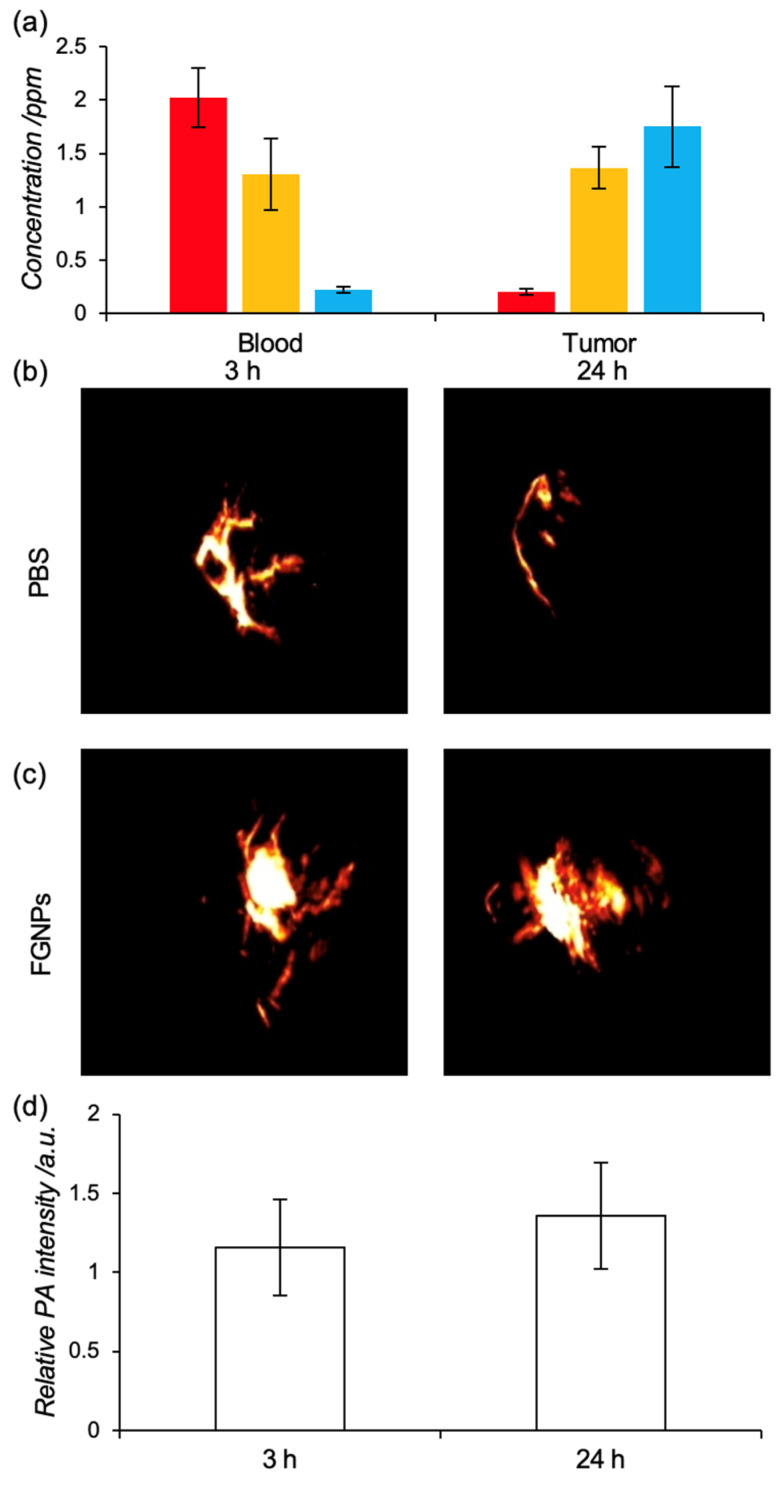
Tumor visualization using FGNPs by photoacoustic imaging. (**a**) Tumor accumulation and blood circulation of FGNPs in tumor xenograft mice (red, 3 h; yellow, 6 h; blue, 24 h). Data represent mean ± SD (*n* = 3). FGNPs were administrated to tumor xenograft mice via intravenous injection. The accumulation of gold was quantified with ICP-AES. (**b**,**c**) Representative photoacoustic images of tumor tissues at 3 and 24 h after injection of PBS (**b**) and FGNPs (**c**) (FGNPs, 300 ppm). (**d**) Photoacoustic signals in tumor tissues. Data represent mean ± SD (*n* = 4).

**Table 1 ijms-23-04686-t001:** Solution properties of FNCs, GNPs, and FGNPs.

	*D_hy_*/nm ^a^	PDI ^a^	ζ-Potential /mV ^b^	[C_60_]/mM	[Au]/ppm ^c^
FNCs	84 ± 5	0.06	−23.1 ± 2.3	1.0	-
GNPs	13 ± 1	0.04	−28.1 ± 3.8	-	300
FGNPs	98 ± 5	0.15	−12.1 ± 4.2	1.0	313

^a^ Hydrodynamic diameter was measured by dynamic light scattering measurement and polydispersity index was calculated by cumulant method. ^b^ ζ-potential was measured by capillary cell and the measurement was carried out in MilliQ (pH, 7.4; 25 °C). ^c^ Concentration of gold was quantified with ICP-AES.

## Data Availability

Not applicable.

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
