# Peer review of "Theranostic Agent Combining Fullerene Nanocrystals and Gold Nanoparticles for Photoacoustic Imaging and Photothermal Therapy"

_ijms, 2022, doi:10.3390/ijms23094686_

Round 1
Reviewer 1 Report
Riku Kawasaki and co-authors have demonstrated the investigation of nanoparticles for diagnosis and therapy of cancer. The aim of research is significant and has prospects for oncological disease cure.
There are some mistakes and difficult understanding moments:
- Could you precise of cancer type in Introduction section (lines 38-43)? I think PTT and PDT methods are not convenient for all cancer types.
- Lines 52-53, what is the aim of sentence? I suppose, that you wanted mentioned biological window transparency in NIR region and its advantages.
- Lines 86-93, formulation of FNCs probably should move to Materials and Methods section.
- There are no legends on fig. 2 (a, c, f).
- In the Introduction section you noticed of infrared regions, but absorption spectra of samples (fig.2a) demonstrated the absence of absorption peaks in NIR region. Are you sure it is correct to mention NIR region in Introduction section?
- Line 108, you presents the particle size and gives the reference to table 1, but there is no table 1 in the Manuscript.
- Line 152, why has you chose laser irradiation at 680 nm? It is confusing because there are no samples absorption at 680 nm.
- Lines 191-192, what method was used for tumor size measurement?
- Materials and Methods section is appropriate to put before Results and Discussion section for the reason that it has coherence and cohesion of the Manuscript and avoid the duplication of equipment characteristics.
Author Response
Referee1:
Comments: Riku Kawasaki and co-authors have demonstrated the investigation of nanoparticles for diagnosis and therapy of cancer. The aim of research is significant and has prospects for oncological disease cure.
Comment 1. Could you precise of cancer type in Introduction section (lines 38-43)? I think PTT and PDT methods are not convenient for all cancer types.
Response to Comment 1: To emphasize advantageous point of PTT and PDT, we added statements in main text as follows:
Page 1, Line 43: and efficient therapeutic efficacy based on photothermal therapy (PTT)[3] and photodynamic therapy toward solid tumor (PDT).[4]
Comment 2. Lines 52-53, what is the aim of sentence? I suppose, that you wanted mentioned biological window transparency in NIR region and its advantages.
Response to Comment 2: We would like to express gratitude to reviewers’ helpful comment to develop understanding of readers. As reviewer supposed, we wanted mention biological window transparency in NIR. In accordance with reviewers’ we revised main text as follows:
Page 2, Line 52: With excellent absorbability in visible light (>600 nm) and near infrared region, which can reach deeper tissues in the body, nanocarbons including carbon nanotubes,[14] and graphene[17] have been used for PAI.
Comment 3. Lines 86-93, formulation of FNCs probably should move to Materials and Methods section.
Response to Comment 3: To avoid duplication in preparation of FNCs, we moved the sentences to Materials and Methods section.
Comment 4. There are no legends on fig. 2 (a, c, f).
Response to Comment 4: To improve readers’ understanding, we added legends on Fig. 2 as follows:
Comment 5. In the Introduction section you noticed of infrared regions, but absorption spectra of samples (fig.2a) demonstrated the absence of absorption peaks in NIR region. Are you sure it is correct to mention NIR region in Introduction section?
Response to Comment 5: Thank you for reviewers’ helpful suggestion. We want to include red light region which is also deeply penetrable in body. In accordance with reviewers’ comment, we added statements as follows:
Page 2, Line 52: With excellent absorbability in visible light (>600 nm) and near infrared region, which can reach deeper tissues in the body, nanocarbons including carbon nanotubes,[14] and graphene[17] have been used for PAI.
Comment 6. Line 108, you presents the particle size and gives the reference to table 1, but there is no table 1 in the Manuscript.
Response to Comment 6: We would like to express gratitude to reviewers’ helpful comment for my carelessness. We added table 1 to the manuscript as follows:
Comment 7. Line 152, why has you chose laser irradiation at 680 nm? It is confusing because there are no samples absorption at 680 nm.
Response to Comment 7: As reviewer pointed out, no obvious peaks are not found in all the samples but only FGNPs have absorbance at 680 nm, which can deeply penetrate in body. Then, we chose the wavelength at 680 nm. In addition, photoacoustic signals and photothermal conversion can be attained efficiently in case of FGNPs, suggesting our system can be activatable by irradiation with laser at 680 nm.
Comment 8. Lines 191-192, what method was used for tumor size measurement?
Response to Comment 8: We used digital caliper to measure the tumor size and tumor volume was calculated by following equation.
Tumor volume = (long axis of tumor) × (short axis of tumor)2 / 2
In accordance with reviewers’ comment, we added following statements in Materials and Method section.
Page 4, Line 157:
Biodistribution of FGNPs Tumor xenograft model mouse were established by subcutaneous injection of suspension of Colon26 cells (5.0 × 105 cells/50 μL). After 10 days incubation, dispersion of FGNPs in PBS were administrated via intravenous injection (10 ppm, 100 μL). Tumor tissue and blood was isolated at each time point (3, 6, and 24 h) and collected organs were digested with aqua regia. The resulting solution was analyzed by ICP-AES. Here, the tumor volume was calculated by following equation.
Tumor volume = (long axis of tumor) × (short axis of tumor)2 / 2
Comment 9. Materials and Methods section is appropriate to put before Results and Discussion section for the reason that it has coherence and cohesion of the Manuscript and avoid the duplication of equipment characteristics.
Response to Comment 9: We would like to express gratitude to reviewers’ helpful comment. We put the Materials and Methods section before Results and Discussion.

Reviewer 2 Report
The authors present the paper "Theranostic Agent Combining Fullerene Nanocrystals and Gold Nanoparticles for Photoacoustic Imaging and Photothermal Therapy". Thank you for so interesting work.
1) Some more 2-3 years references are required for the Introduction section. Also, the review papers are more preferred.
2) Figure S3. What DLS mode have you used? Can you mention it in SI Figure S3. DLS spectra of nanoparticles in size and intensity mode have to be presented. It can be an additional characteristic of nanoparticles besides TEM. The clear calculation of PDI and size can be presented in the main part of the paper.
3) What does aqueous media mean for Fig. S3. I haven't seen clear conditions (concentration, pH, salt, etc.) for Figure S3.
4) You have two S3 Figures in SI, but no S4.
Cultural media stability is a very good experiment, but it is not mimicking the bloodstream. There are a lot of salts but not the same amount of proteins such as albumin, etc. So I think you need to do a stability experiment in blood plasma or rewrite the pictures caption and text in the paper.
"In case of FGNPs, the hydrodynamic diameter gradually increased from 98 nm to 150 nm with incubation time, indicating absorption of serum proteins resulted in secondary aggregation of FGNPs". After this sentence, it looks like your nanoparticles are not stable in organism conditions. Can you comment on it in the paper? I think you need a much more extensive discussion of the construction stability.
5) In the conclusion section, some future perspectives and limitations of the work have to be presented.
Minor comments
1) cell viability. It is relative. In this way, can you mention the control solution somewhere? Usually the cells with some buffer.
2) Fig 3c. What do the colors mean? Fig 3d what does black mean?
3) 'Cell destruction was dose-dependently induced to the cells in all systems (Figure 3c and Figure S4).' Maybe, Fig S5? In Fig S5 what do colors mean?
Author Response
Referee: 2
Comments: The authors present the paper "Theranostic Agent Combining Fullerene Nanocrystals and Gold Nanoparticles for Photoacoustic Imaging and Photothermal Therapy". Thank you for so interesting work.
Comment 1: Some more 2-3 years references are required for the Introduction section. Also, the review papers are more preferred.
Response to Comment 1: We would like to express our gratitude to reviewers’ valuable comment to improve our paper. In accordance with reviewers’ comment, we included following papers as references and revised the number for the references.
- Cook, A.B.; Decuzzi, P.; Harnessing Endogenous Stimuli for Responsive Materials in Theranostics, ACS Nano 2021, 15, 2068-2098.
- Malavika, J. P.; Shobana, C.; Sunbagamoorthy, S.; Ganeshbabu, M.; Kumar, P.; Selvan, R. K. Green Synthesis of Multifunctional Carbon Quantum Dots: An Approach in Cancer Theranostics. Biomater. Adv. in press (DOI: 10.1016/j.bioadv.2022.212756)
Comment2: Figure S3. What DLS mode have you used? Can you mention it in SI Figure S3. DLS spectra of nanoparticles in size and intensity mode have to be presented. It can be an additional characteristic of nanoparticles besides TEM. The clear calculation of PDI and size can be presented in the main part of the paper.
Response to Comment 2 : We would like to express our gratitude to reviewers’ helpful comment to improve our paper. In this work, we employed data from intensity mode to evaluate size of the nanoparticles because the data is the most reliable. In accordance with reviewers’ comment we added Table 1 for presenting solution properties of the nanocrystals including the hydrodynamic diameter, size distribution, zeta-potential. In addition, we added DLS spectra in Supplemental Information as follows:
Comment 3. What does aqueous media mean for Fig. S3. I haven't seen clear conditions (concentration, pH, salt, etc.) for Figure S3.
Response to Comment 3: To clear the condition for the measurement, we revised the caption as follows:
Comment 4. You have two S3 Figures in SI, but no S4.
Response to Comment 4 : We would like to express gratitude to reviewers’ comment and I apologies to my carelessness. I duplicated Figure S3 and S4, then we replaced the Figure S4 to correct one.
Comment 5. Cultural media stability is a very good experiment, but it is not mimicking the bloodstream. There are a lot of salts but not the same amount of proteins such as albumin, etc. So I think you need to do a stability experiment in blood plasma or rewrite the pictures caption and text in the paper. "In case of FGNPs, the hydrodynamic diameter gradually increased from 98 nm to 150 nm with incubation time, indicating absorption of serum proteins resulted in secondary aggregation of FGNPs". After this sentence, it looks like your nanoparticles are not stable in organism conditions. Can you comment on it in the paper? I think you need a much more extensive discussion of the construction stability.
Response to Comment 5: Thank you for the valuable comment. For the point of view to reduce animal experiments in animal welfare, we revised manuscript as follows and added statements:
Page 6, Line 229: Though FGNPs formed secondary aggregation in culture media condition, their hydrodynamic diameter did not change after the formulation of secondary aggregation with maintaining the particle size corresponding to EPR effect, suggesting that our system can be potentially applicable for contrast agent for tumor visualization.
Comment 6. In the conclusion section, some future perspectives and limitations of the work have to be presented
Response to Comment 6: In accordance with reviewers’ suggestion, we added future perspective and limitations of the work as follows:
Page 9, Line 303: Absorbability of our system might be insufficient to visualize specific organ and disease for the absorption in visible light is partially overlapped with the heme proteins such as myoglobin and hemoglobin. For the point of view, ripening of gold nanorod on the surface of FNCs can exploit the absorbability in NIR region. Moreover, modification of pilot molecules on the surface of FGNPs open the opportunity for the application to visualize specific disease with high sensitivity.
Comment 7. Cell viability. It is relative. In this way, can you mention the control solution somewhere? Usually the cells with some buffer.
Response to Comment 7: To mention the control solution, we added statements in Materials and Methods section as follows:
Page 4, Line 138: In the control, we added same volume of milliQ (1 μL), which is used to prepare samples, to the cells.
Comment 8. Fig 3c. What do the colors mean? Fig 3d what does black mean?
Response to Comment 8: We added explanation for Fig. 3c. In Fig. 3d, black means photoacoustic signals from the control cells without any treatment. To clarify what do the color indicate, we revised caption in Figure 3 as follows:
Figure 3 Caption: (c) Photo-induced cytotoxicity toward Colon26 cells using FGNPs (blue) and GNPs (red). The cells were co-incubated with FGNPs with varying concentration for 24 h. The cells were exposed to light with optimal wavelength (>620 nm) for 30 mins. After 24 h incubation, the cell viability was confirmed by Cell Counting Kit-8 (n=3). Data represents mean±SD. (d) Photoacoustic signal from Colon26 cells treated with FGNPs (blue). Colon26 cells were treated with FGNPs (20 ppm, blue) orMilliQ (nontreatment control, black) for 24 h and the collected cells were excited at 680 nm (n = 3). Data represents mean±SD.
Comment 9. 'Cell destruction was dose-dependently induced to the cells in all systems (Figure 3c and Figure S4).' Maybe, Fig S5? In Fig S5 what do colors mean?
Response to Comment 9: We would like to express our gratitude to reviewers’ helpful suggestion and I apologies my careless. As reviewer pointed out, the Figure S4 is Figure S5. In accordance with reviewers’ comment, we revised manuscript and Supporting Information as follows:
Page 8, Line 268: To address application for PTT, photoirradiation (>620 nm) were carried out to Colon26 cells treated with FNCs, FGNPs or GNPs. Cell destruction was dose dependently induced to the cells in FGNPs and GNPs (Figure 3c) but FNCs could not kill cancer cells even at the highest concentration (1 mM of C60) (Figure S5).

Round 2
Reviewer 1 Report
Thank you for the significant research. Now the manuscript is fine.
Reviewer 2 Report
Thank you for so high quality research. I will look forward on your next work.